# Investigating the stratified efficacy and safety of pharmacological blood pressure-lowering: an overall protocol for individual patient-level data meta-analyses of over 300 000 randomised participants in the new phase of the Blood Pressure Lowering Treatment Trialists' Collaboration (BPLTTC)

Kazem Rahimi,[1] Dexter Canoy,[1] Milad Nazarzadeh,[1,2]
Gholamreza Salimi-Khorshidi,[1] Mark Woodward,[3] Koon Teo,[4] Barry R Davis,[5]
John Chalmers,[6] Carl J Pepine,[7] on behalf of the Blood Pressure Lowering
Treatment Trialists' Collaboration

For numbered affiliations see end of article.

**Correspondence to**
Professor Kazem Rahimi;
kazem.rahimi@georgeinstitute.ox.ac.uk

## ABSTRACT

**Introduction** Previous research from the Blood Pressure Lowering Treatment Trialists' Collaboration (BPLTTC) and others has shown that pharmacological blood pressure (BP)- lowering substantially reduces the risk of major cardiovascular events, including ischaemic heart disease, heart failure and stroke. In this new phase, the aim is to conduct individual patient-level data (IPD) meta-analyses involving eligible BP-lowering randomised controlled trials (RCTs) to address uncertainties relating to efficacy and safety of BP-lowering treatment.

**Methods and analysis** RCTs investigating the effect of pharmacological BP-lowering, with a minimum of 1000 patient-years of follow-up in each trial arm, are eligible. Our systematic review identified 100 potentially eligible trials. We requested their investigators/sponsors to contribute baseline, follow-up and outcomes data. As of June 2018, the collaboration has obtained data from 49 trials (n=315 046 participants), with additional data currently in the process of being transferred from four RCTs (n=34 642 participants). In addition, data harmonisation has commenced. Scientific activities of the collaboration are overseen by the Steering Committee with input from all collaborators. Detailed protocols for individual meta-analyses will be developed and registered on public platforms.

**Ethics and dissemination** Ethics approval has been obtained for this new and extended phase of the BPLTTC, the largest collaboration of de-identified IPD from RCTs. It offers an efficient and ethical manner of re-purposing existing data to answer clinically important questions relating to BP treatment as well as methodological questions relating to IPD meta-analyses. Among the immediate impacts will include reliable quantification of

### Strengths and limitations of this study

► This protocol aims to conduct individual patient-level data (IPD) of blood pressure (BP)-lowering randomised clinical trials (RCT) to address stratified efficacy of BP-lowering treatment according to various patient characteristics, examine the impact of BP-lowering treatment on less-studied vascular outcomes, and identify unintended consequences of such treatment.

► With data from 49 RCTs (>300 000 participants) already shared in the collaboration, it will be the largest IPD meta-analysis of any randomised trials, allowing sufficient statistical power to address a number of important clinical questions a single BP-lowering randomised trial is unlikely to be able to answer reliably on its own.

► Findings of meta-analyses from this resource are likely to provide reliable evidence to inform clinical guidelines and public health policy development.

► Data from some RCTs are not accessible because some investigators, particularly of older trials, could not be traced or sponsors/funders of the trials are unwilling to share data.

► Not all RCTs would have collected the same clinical information so missing data are inevitable for some studies; appropriate methods to address these issues would have to be utilised.

effects of treatment modifiers, such as baseline BP, age and prior disease, on both vascular and non-vascular outcomes. Analyses will further assess the impact of BP-lowering on important, but less well understood, outcomes,

such as new-onset diabetes and renal disease. Findings will be published in peer-reviewed medical journals on behalf of the collaboration.

## INTRODUCTION

Elevated blood pressure (BP) is a major risk factor for cardiovascular death and disability worldwide, and its burden has been increasing. In 2016, elevated BP above the minimum risk exposure threshold of 110 mm Hg systolic was responsible for >10 million deaths and 7% of the world's total disability-adjusted life years .[1]

Several randomised trials and their meta-analyses have shown that pharmacological BP-lowering is an effective and affordable strategy for management of cardiovascular risk. Despite this, treatment and control rates among those who have been classified to have 'hypertension' (BP >140/90 mm Hg) remain low in both high-income and low-income countries.[2] This gap between evidence and practice is likely to be at least in part due to remaining uncertainties about the efficacy and safety of BP-lowering.[3] For instance, there is no expert consensus on the appropriate BP threshold for treatment initiation and whether this should differ by age or disease status (eg, diabetes or coronary heart disease (CHD)). Also, little is known on the effects of BP-lowering on safety outcomes (eg, cancer, falls and fractures, and acute kidney injury), non-vascular outcomes (eg, new-onset diabetes, dementia) or less commonly reported vascular outcomes (eg, atrial fibrillation, peripheral arterial disease, aortic aneurysms, pulmonary embolism and stroke subtype).

These existing uncertainties have major consequences for patient management and outcomes. On the one hand, lack of reliable estimates for the efficacy of BP lowering on less commonly reported vascular outcomes, some of which have a clear association with BP in observational studies,[4–7] may result in an underestimation of the overall impact of BP-lowering in the general patient population. On the other hand, uncertainty about the safety of BP-lowering could potentially expose specific patient groups to avoidable harms because of overuse or underuse of BP-lowering treatment. To address these uncertainties, more evidence from randomised trials are needed. However, because single randomised trials are often not large enough for detailed investigation of the safety and efficacy of BP-lowering and its variation by a class of medication or important patient features, pooling information from several trials remains the most efficient and often the only viable option for addressing these remaining uncertainties.

In 1995, principal investigators of major ongoing clinical trials of BP-lowering drugs conceived and initiated the Blood Pressure Lowering Treatment Trialists' Collaboration (BPLTTC) to investigate the effect of BP-lowering on major cardiovascular events and all-cause mortality through a series of individual patient-level data (IPD) meta-analyses.[8] Since its inception, the BPLTTC has successfully completed several studies which have provided reliable evidence for the benefits of BP-lowering

for prevention of major cardiovascular outcomes (CHD, heart failure and stroke) and have demonstrated that major classes of antihypertensives have similar beneficial effects on major vascular outcomes.[9–13] However, with its limited size and original focus on major cardiovascular outcomes only, the BPLTTC has been unable to address important contemporary questions about the stratified (ie, subgroup-specific) effects of BP-lowering and its safety.

## OBJECTIVES

In 2014, existing BPLTTC investigators were approached and asked to extend the scope of the collaboration with the provision of information on all available and accessible baseline, follow-up and outcomes data. In addition, investigators from several other large-scale trials who had not participated in BPLTTC or had provided summary data only were invited to join the collaboration with the provision of IPD. The key objectives for this extended phase of BPLTTC were specified as follows:

1. To assess the stratified efficacy of BP-lowering on major vascular events (CHD, stroke and heart failure) and all-cause mortality by baseline BP levels, sex, age, disease status (including multimorbidity) and risk of CVD at baseline.
2. To assess the overall and stratified safety of BP-lowering by examining its effect on unintended outcomes, including total serious adverse events, acute renal failure, new-onset diabetes, fractures and cancer overall, and by drug classes and baseline BP levels.
3. To assess the extended efficacy of BP-lowering on other important 'vascular' outcomes, including atrial fibrillation and dementia.

## METHODS AND ANALYSIS
### Inclusion and exclusion criteria

For this extended phase of the collaboration, the original trial eligibility criteria were modified, chiefly to enable inclusion of additional large-scale trials that were reported before the inception of the BPLTTC in 1995. We will only include randomised clinical trials to minimise the impact of bias and confounding. For clinical trials to be eligible for inclusion in BPLTTC, one of the following criteria must be met:

▶ Randomisation of patients between a BP-lowering agent and a placebo arm (or other inactive control condition).
▶ Randomisation of patients between various BP-lowering intensities.
▶ Randomisation of patients between various antihypertensive drugs.

Additionally, a minimum of 1000 patient-years of follow-up in each randomly allocated trial arm is required. There is no restriction on publication date, setting or drugs investigated. However, the following types of trial will be excluded:

- ► Trials exclusively conducted in patients with heart failure or short-term interventions in patients with acute myocardial infarction or other acute settings.
- ► Trials with non-pharmacological interventions of BP-lowering without a drug comparison arm.
- ► Trials without a clearly defined randomisation process.

## Identification of trials

We conducted a systematic review to identify trials for possible inclusion in the IPD meta-analysis. The search was restricted to clinical trials, controlled clinical trials, randomised controlled trials or meta-analyses and no language restrictions were applied. We searched electronic bibliographic databases, including PubMed/Medline (NCBI, Bethesda, MD, USA), The Cochrane Central Register of Controlled Trials (The Cochrane Collaboration, London, UK) and the ClinicalTrials.gov website covering the periods between 1 January 1966 and 1 June 2018. Reference lists of eligible studies, related meta-analyses, clinical trial registries and BPLTTC collected in previous projects[14] were hand-searched to identify further relevant studies. This systematic review protocol, including details of the methods and search strategy, was registered with PROSPERO (www.crd.york.ac.uk/prospero/) (CRD42018099283) before the review was conducted. The online supplementary table 1 shows the Medline search strategy. For this protocol, the Preferred Reporting Items for Systematic review and Meta-Analysis Protocols (PRISMA-P) checklist is shown in the online supplementary table 2.

We used reference management software (Endnote) to store identified records from different sources. Duplicate records from various sources were collapsed into a single entry. Two investigators (DC and MN) conducted independent searches and screened the title and abstracts of publications for meeting eligibility criteria. The full manuscript was then requested and assessed for eligibility. Disagreements were resolved by discussion, with the involvement of a third investigator (KR) where necessary.

Because the purpose of this systematic review was to identify potential trials for inclusion in an IPD meta-analysis, we mainly extracted information about the investigators and/or contact details to allow us to invite them to participate by contributing data. When conducting the meta-analysis of RCTs for which we have obtained data at individual patient-level, we will report quality assessment of these RCTs based on randomisation procedures, treatment allocations according to assignment, and outcomes collected and compared across comparison groups; concealment and blinding methods; and risk of bias both at study and outcome levels using the Cochrane Risk of Bias Tool. We will use the quality of RCTs as part of sensitivity analysis when conducting the IPD meta-analysis.

Figure 1 summarises the trial search as of June 2018, indicating that a total of 100 trials with 595 514 randomised participants were identified which potentially met the BPLTTC eligibility criteria (online supplementary table

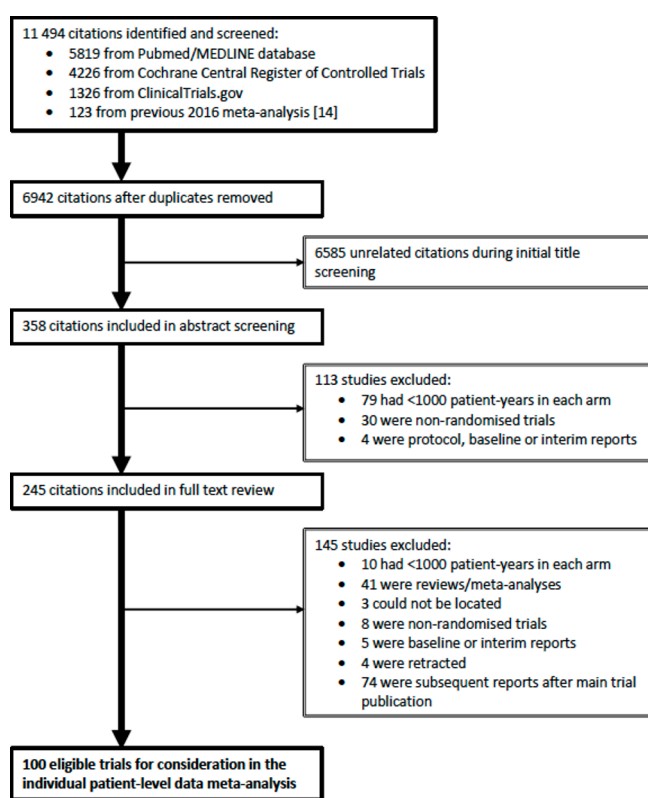

**Figure 1** Literature search and trial selection flowchart.

3). This literature search will be continuously updated, and investigators of newly reported and eligible trials will be invited to join the collaboration.

## DATA COLLECTION

All existing BPLTT collaborators and investigators of all newly identified trials were contacted and asked whether they would be willing to provide IPD (variables are listed in the online supplementary table 4). As of June 2018, the collaboration had acquired data from 49 trials comprising 315 046 participants. Investigators or sponsors of several other trials have agreed to share data. Data from another four major trials with 34 642 participants are currently at the transfer stage. For the remaining trials, data are still being sought. Many of these trials have been published many years ago and identification of the data guardian or an electronic database of the trial has proven challenging.

In parallel to requesting further data, the process of data harmonisation has commenced. This includes a creation of a database with all information obtained from investigators, using the data dictionary provided. Initial data verification will include checking of the individual patient data with respect to a number of participants, baseline characteristics, and types of follow-up and outcome data available. Next, prepared data are to be checked for consistency (eg, plausible ranges, randomisation sequence and so on) and completeness. In case of any inconsistencies or missing information, the investigators will be contacted for verification.

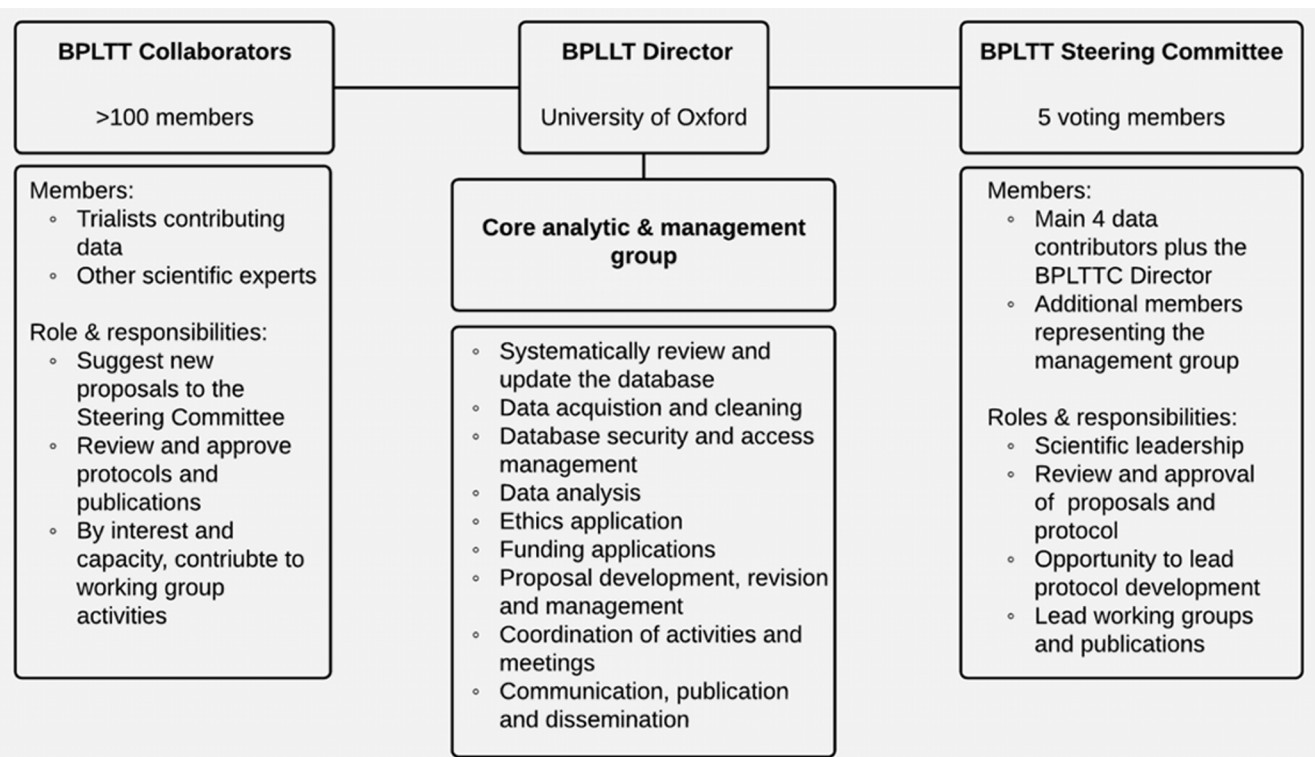

**Figure 2** Overview of collaboration governance. BPLTT, Blood Pressure-Lowering Treatment Trialists' Collaboration

## Governance and data management policies

Figure 2 provides an overview of the roles and responsibilities of the BPLTTC collaborative group. The BPLTTC Director, supported by a core analytic and management group, at the University of Oxford coordinates all activities among internal collaborators and external stakeholders. This central team will periodically update the literature review, acquire new data, develop proposals and protocols, seek approvals and funding, and manage publications and communication. Scientific activities of the BPLTTC core group are overseen by the Steering Committee with input from all BPLTT collaborators.

The aforementioned three main objectives of the extended phase of BPLTTC have all been approved by the Steering Committee and collaborators. For each of these objectives, at least one study protocol will be developed. Collaborators or external researchers may suggest additional research proposals to the BPLTTC core group. Such proposals shall be checked for feasibility and overlap with any existing proposals. If appropriate, they will be shared with the Steering Committee for scientific review. After approval, proposing researchers will be able to join a working group, including other interested members of the Collaboration. This working group will work with the BPLTTC core group and Steering Committee to write the research protocol, conduct and interpret the results, draft the manuscript and publish the final report.

Once agreement on a structured protocol is reached by the working group, the study will be registered with PROSPERO and the protocol will be posted on a password protected website at www.bplttc.org. Additionally, collaborators will be able to use this website to stay informed about the progress of various projects.

Trial data from investigators and sponsors are typically transferred by using a secure file transmission system, such as Oxford's Oxfile system. Before sending a data file, the data will be compressed using a standard compression system and then encrypted. All data transfer processes will be overseen by the BPLTTC core group in compliance with the data exchange and sharing agreement with the data provider. All trial data will be considered confidential and will not be provided to any third party without written permission of the owner of the data. Data will be stored on site at The George Institute at the University of Oxford, with strict confidentiality and comprehensive data security. All analyses will be carried out by the core analytic team, according to research priorities and availability of resources.

## Patient and public involvement

This protocol describes plans for collaborative research investigations involving secondary analyses of existing clinical trials data. There was neither any involvement by patients in conceptualising the study nor in the development of the protocol.

## Statistical analysis

Prior to commencing any analysis, a feasibility assessment and power calculation will be carried out on the blinded dataset. For approved proposals, a protocol will be written which will entail a detailed statistical analysis plan. Although specific proposals may require different

statistical methods, in principle, we intend to use a one-step approach by simultaneously analysing observations from all trials (while accounting for clustering of observations within studies).[15 16] This approach is more suitable than a two-step approach when non-linear effects or multiple covariates and their interaction are to be assessed, which our stratified analyses aim to do.[17] For some analyses, it might be important to supplement the available IPD with aggregate data for those studies where IPD are not available, using a combined one-step and two-step approaches. Aggregate data will be obtained by sending our computer codes to the Investigators of such studies, so as to extract their results (eg, HRs), rather than their data.

We will use fixed effects methods to pool trial-specific data. For time-to-event data, clustered Cox proportional-hazard models will be used and HRs with 95% or 99% CIs will be reported. For binary outcomes, risk ratios and 95% CI will be estimated using log-binomial mixed-effects models (or ORs from logistic models, if necessary). Effects on continuous outcomes (eg, renal function) will be quantified by mean differences and 95% CI and will be estimated using linear mixed effects models.

Using a two-step approach, key results will be presented using Forest plots, and Cochran's Q test and $I^2$ statistic will be used to quantify the degree of heterogeneity between studies. Meta-regression will be used if considerable residual heterogeneity remains after controlling for all possible variables and recognised effect modifiers, with P-value adjusted for false positive findings based on Monte Carlo simulation.

Where appropriate, analyses will also be stratified by class of medication, patient population and baseline BP to allow for heterogeneity among different patient populations and types of medication to be determined. In addition, a dose-effect (dose-response) regression model will be applied to assess any potential dose-effect association for each medication dosage and represent efficacy and probable risk of side effects (point-to-point association) both at individual and trial levels (aggregate data). Also, we will use fractional polynomials and cubic splines to smoothen the dose–response relationship and address any violated linearity assumption.[18] For analysing drug class-specific effects, we will consider using IPD network meta-analysis.[19]

Most trials included in BPLTTC have a double-blind design where the use of routine unadjudicated outcomes is unlikely to result in biased estimates (provided a sufficient number of events can be gathered) because any under- reporting or over-reporting would affect both arms of the trial equally.[20 21] However, due to the different data formats of contributing trials and potentially different types of data provided, additional methodological work might be needed to enhance the quality and power of the data. This might be particularly relevant when a large proportion of data on important covariates, effect modifiers or outcomes is missing. Simply ignoring the missing data, or implementing a complete case analysis, is typically inappropriate, as it could result in biased results. We will, therefore, conduct methodological studies that will compare existing and novel methods for imputation of non-commensurate measurements between trials, correction of measurement errors or harmonisation of differing definitions among trials.[22–25]

In general, reporting of the findings will follow the PRISMA-IPD guidance[26] and where appropriate, we will use Grading of Recommendations Assessment, Development and Evaluation (GRADE) (www.gradeworkinggroup.org) when assessing the certainty of evidence.

## Ethics and dissemination

As we do not have access to patient identifiable information, our procedures are fully compliant with the Data Protection Act 1998 and consistent with the principles of the General Data Protection Regulation. The University of Oxford's research contracts team is well aware of legal requirements for data sharing and storage and all our procedures are fully compliant with these requirements. These requirements are described in detail in Oxford's research governance (https://researchsupport.admin.ox.ac.uk/governance/integrity) and policies on research data handling and management (http://researchdata.ox.ac.uk/). We have already obtained ethics review from the Oxford Central Ethics University Research Committee (OxTREC Reference: 545–14) and agreed to communicate any major amendments to the protocol for this third cycle of BPLTTC with the Ethics Committee for further approval before analyses begin.

The knowledge of the degree of efficacy and safety of BP-lowering and its variation by a class of medication and among subgroups of patients will help to refine treatment recommendations. The presence of clinically meaningful differences in the efficacy of different classes of BP medication, either overall, or within certain patient groups will have considerable implications for stratified patient management and clinical decision-making. BPLTTC is the largest international hypertension collaboration and is unique in its ability to pool and analyse IPD from a large number of trials to answer these clinically relevant questions.

We will publish our findings in peer-reviewed medical journals. Further, we will share and present our findings at various scientific meetings and through the Collaboration's networks and memberships across professional societies. The George Institute for Global Health has communication and dissemination practices in place to ensure that research it produces and policies it contributes to are regularly transmitted through its various institutes and units in Australia, India, China, UK and the USA and to its partner organisations globally.

Details about data curation and deposition are described under the section on 'Governance and data management policies'. No data will be provided to any third party without written permission from the trial data provider or custodian. As previously mentioned, the BPLTTC is governed by the University of Oxford's policies on research integrity

and codes of practice and follow the university policy on the management of research data and records.

## Author affiliations

[1]The George Institute for Global Health, University of Oxford, Oxford, Oxfordshire, UK
[2]Collaboration Centre of Meta-Analysis Research, Torbat Heydariyeh University of Medical Sciences, Torbat Heydariyeh, Iran
[3]University of New South Wales, Sydney, New South Wales, Australia
[4]Population Health Research Institute, Hamilton, Ontario, Canada
[5]University of Texas Health Science Center, Houston, Texas, USA
[6]The George Institute for Global Health, Sydney, New South Wales, Australia
[7]College of Medicine, University of Florida, Gainesville, Florida, USA

**Collaborators** The Blood Pressure Lowering Treatment Trialists' Collaboration: Steering Committee: Kazem Rahimi (Chair) (The George Institute for Global Health, University of Oxford, Oxford, United Kingdom), Koon Teo (Population Health Research Institute, McMaster University, Hamilton, Ontario, Canada) Barry R Davis (The University of Texas School of Public Health, Houston, Texas, USA), John Chalmers (The George Institute for Global Health, University of New South Wales, Sydney, Australia), Carl J. Pepine (Department of Medicine, University of Florida, Gainesville, Florida, USA). Collaborating Trialists: L Agodoa (AASK [African-American Study of Kidney Disease and Hypertension]), A Algra (Dutch TIA Study [Dutch Transient Ischemic Attack Study]), F W Asselbergs (PREVEND-IT [Prevention of Renal and Vascular End- stage Disease Intervention Trial]), N Beckett (HYVET [Hypertension in the Very Elderly Trial]), E Berge (VALUE trial [Valsartan Antihypertensive Long-term Use Evaluation trial]), H Black (CONVINCE [Controlled Onset Verapamil Investigation of Cardiovascular End Points]), F.P.J. Brouwers (PREVEND-IT), M Brown (INSIGHT [International Nifedipine GITS Study: Intervention as a Goal in Hypertension]), C J Bulpitt (HYVET), B Byington (PREVENT [Prospective Randomized Evaluation of the Vascular Effects of Norvasc Trial]), J Chalmers (ADVANCE [Action in Diabetes and Vascular Disease: Preterax and Diamicron MR Controlled Evaluation]), J Cutler (ALLHAT [Antihypertensive and Lipid-Lowering Treatment to Prevent Heart Attack Trial]), B Davis (ALLHAT), R B Devereaux (LIFE [Losartan Intervention For Endpoint reduction in hypertension]), D Dwyer (IDNT [Irbesartan Diabetic Nephropathy Trial]), R Estacio (ABCD [Appropriate Blood Pressure Control in Diabetes]), R Fagard (SYST-EUR [SYSTolic Hypertension in EURope]), K Fox (EUROPA [European trial on Reduction Of cardiac events with Perindopril among patients with stable coronary Artery disease]), T Fukui (CASE-J [Candesartan Antihypertensive Survival Evaluation in Japan]), A J Gupta (ASCOT [Anglo-Scandinavian Cardiac Outcomes Trial]), R R Holman (UKPDS [UK Prospective Diabetes Study]), Y Imai (HOMED-BP [Hypertension Objective Treatment Based on Measurement by Electrical Devices of Blood Pressure]), M Ishii (JMIC-B [Japan Multicenter Investigation for Cardiovascular Diseases-B]), S Julius (VALUE), Y Kanno (E-COST [Efficacy of Candesartan on Outcome in Saitama Trial]), S E Kjeldsen (VALUE, LIFE), J Kostis (SHEP [Systolic Hypertension in the Elderly Program]) K Kuramoto (NICS-EH [National Intervention Cooperative Study in Elderly Hypertensives]), J Lanke (STOP2 [Swedish Trial in Old Patients with Hypertension-2], NORDIL [Nordic Diltiazem]), E Lewis (IDNT), J Lewis (IDNT) M Lievre (DIABHYCAR [Non-insulin-dependent diabetes, hypertension, microalbuminuria or proteinuria, cardiovascular events, and ramipril study]), L H Lindholm (CAPPP [Captopril Prevention Project], STOP2, NORDIL), S Lueders (MOSES [The Morbidity and Mortality After Stroke, Eprosartan Compared With Nitrendipine for Secondary Prevention]), S MacMahon (ADVANCE), M Matsuzaki (COPE [The Combination Therapy of Hypertension to Prevent Cardiovascular Events]), M H Mehlum (VALUE), S Nissen (CAMELOT [Comparison of Amlodipine vs Enalapril to Limit Occurrences of Thrombosis]), H Ogawa (HIJ-CREATE [Heart Institute of Japan Candesartan Randomized Trial for Evaluation in Coronary Heart Disease]), T Othisgihara (CASE-J), T Ohkubo (HOMED-BP), C Palmer (INSIGHT), A Patel (ADVANCE), C Pepine (INVEST [International Verapamil SR-Trandolapril Study]), M Pfeffer (PEACE [Prevention of Events With Angiotensin- Converting Enzyme Inhibition]), N R Poulter (ASCOT [Anglo-Scandinavian Cardiac Outcomes Trial]), H Rakugi (VALISH [Valsartan in Elderly Isolated Systolic Hypertension Study], CASE-J), G Remuzzi (BENEDICT [BErgamo NEphrologic Diabetes Complications Trial]), P Ruggenenti (BENEDICT), T Saruta (CASE-J), J Schrader (MOSES), R Schrier (ABCD), P Sever (ASCOT), P Sleight (CONVINCE, HOPE [Heart Outcomes Prevention Evaluation], TRANSCEND [Telmisartan Randomised AssessmeNt Study in ACE iNtolerant subjects with cardiovascular Disease], ONTARGET [Ongoing Telmisartan Alone and in Combination with Ramipril Global Endpoint Trial]), J A Staessen (SYST-EUR [Systolic Hypertension in Europe]), H Suzuki (ECOST), L Thijs (Syst-Eur), K Ueshima (VALISH, CASE-J), S Umemoto (COPE), W H van Gilst (PREVEND-IT), P Verdecchia (Cardio-Sis [CARDIOvascolari del Controllo della Pressione Arteriosa SIStolica]), K Wachtell (LIFE), L Wing (ANBP2 [The Second Australian National Blood Pressure Study]), Y Yui (JMIC-B), S Yusuf (HOPE, ONTARGET, TRANSCEND), A Zanchetti (deceased) (VHAS [Verapamil in Hypertension and Atherosclerosis Study], ELSA [European Lacidipine Study on Atherosclerosis]). Other members: C Baigent, R Collins, D de Zeeuw B Neal, V Perkovic, M Rahman, W J Remme, A Rodgers, J Sundström, F Turnbull.

**Contributors** KR: wrote the first draft of the manuscript; KR and DC: sought access to the data; DC, MN and KR: conducted the systematic review; KR, DC, MN, GSK, MW, KT, BRD, JC and CJP: contributed in the conception and design of the study, provided critical input into the subsequent revisions of the manuscript and gave approval to the final version to be published; KR and DC: are guarantors of this protocol.

**Funding** BPLTTC is supported by the Oxford Martin School and the National Institute for Health Research (NIHR) Oxford Biomedical Research Centre (BRC).

**Disclaimer** The views expressed are those of the author(s) and not necessarily those of the NHS, the NIHR or the Department of Health; Funders have no involvement in the conduct of the review or in the development of the protocol.

**Competing interests** George Health Enterprises, the social enterprise arm of The George Institute for Global Health, has received investment to develop fixed-dose combination products containing aspirin, statin, and blood pressure-lowering drugs. George Health Enterprises has submitted patents for low-dose blood pressure combinations on which one of the collaborators (A Rodgers) is listed as one of the inventors, but does not have a financial interest in these planned products.

**Provenance and peer review** Not commissioned; externally peer reviewed.

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
