## [Reviewer comments · BMJ Open]

ARTICLE DETAILS

TITLE (PROVISIONAL)	Investigating the stratified efficacy and safety of pharmacological blood pressure lowering: An overall protocol for individual patient-level data meta-analyses of over 300,000 randomized participants in the new phase of the Blood Pressure Lowering Treatment Trialists' Collaboration (BPLTTC)
AUTHORS	Rahimi, Kazem; Canoy, Dexter; Nazarzadeh, Milad; Khorshidi, Gholamreza; Woodward, Mark; Teo, Koon; Davis, Barry; Chalmers, John; Pepine, Carl

VERSION 1 - REVIEW

REVIEWER	Suzanne Oparil Division of Cardiovascular Disease Department of Medicine, School of Medicine University of Alabama at Birmingham Birmingham, Alabama, USA I have served on the Steering Committees of numerous important outcome trials in hypertension (e.g., ALLHAT, LIFE, Symplicity HTN-3 and SPRINT). I served as Director/PI of the UAB Clinical Center Network (CCN) for the NIH/NHLBI-funded SPRINT trial.
REVIEW RETURNED	15-Jan-2019

GENERAL COMMENTS	This manuscript describes the impressive recent activity of the Blood Pressure Lowering Treatment Trialists' Collaboration (BPLTTC) that aims to conduct individual patient-level data (IPD) meta-analyses of published BP lowering randomized controlled trials (RCTs) in order to further enhance our understanding of the efficacy and safety of BP lowering treatment. The investigators have already made substantial progress in obtaining data from 49 trials with over 315,000 participants in all. The novelty of this approach is that it will yield sufficient IPD to assess the effects of BP lowering as a whole and also BP lowering with specific drug classes on a large variety of clinical outcomes, including, in addition to major cardiovascular events, new-onset diabetes, peripheral vascular disease, atrial fibrillation, fractures, cancer, cognitive function decline and vascular dementia. The adverse effects of antihypertensive treatment are less well appreciated than the positive effects of BP lowering based on individual trial results; the proposed analysis will fill in the gaps unsatisfied by individual trial results and will have major impact on individual patient management, clinical decision making and antihypertensive treatment guidelines. The organization of the
---

	investigation and the quality of the investigators are first rate. There are no major weaknesses in the proposal.
--	---

REVIEWER	Toshi A. Furukawa Kyoto University Graduate School of Medicine / School of Public Health TAF reports personal fees from Meiji, Mitsubishi-Tanabe, MSD and Pfizer and a grant from Mitsubishi-Tanabe, outside the submitted work; TAF has a patent 2018-177688 pending.
REVIEW RETURNED	16-Jan-2019

GENERAL COMMENTS	The authors present an overall protocol (not a specific protocol or statistical analysis plan) of their third round of IPD meta-analyses of blood pressure lowering drugs. They present the study selection procedure (which has already been completed), the data collection and management procedures (which are ongoing), and list their three broad categories of primary outcomes, but they do not present detailed statistical analyses. Instead they write “For each of these objectives, at least one study protocol will be developed.” The author team has a long tradition of conducting meaningful analyses based on IPD and I have only minor comments. MINOR BUT ESSENTIAL POINTS  [ ] Could the authors present more details about the ethical and legal issues involved in collecting, managing and analyzing IPD? [ ] The authors write, “As we do not have access to patient identifiable information, our procedures are fully compliant with the Data Protection Act 1998.” But how about the newer regulations such as EU’s GDPR? And how about obtaining IPD from non-EU countries? Do they present particular difficulties? [ ] The authors also write, “The University of Oxford’s research contracts team is well aware of legal requirements for data sharing and storage and all our procedures are fully compliant with these requirements.” What are “these requirements”? The readers would appreciate more details and difficulties involved in these processes. [ ] The title should reflect the fact that the authors do not report details of analyses for specific outcomes but intend to write separate protocols for them. Otherwise the readers might assume that paper is such a protocol (as is usually the case for a systematic review protocol). Probably “An overall protocol” instead of simply “A protocol”? [ ] In COI statement, the authors write, “George Health Enterprises, the social enterprise arm of The George Institute, has received investment to develop fixed dose combinations of aspirin, statin and blood pressure lowering drugs.” Which “blood pressure lowering drugs”? MINOR AND DISCRETIONARY POINTS  [ ] I would assume that the included trials would involve many different drugs. Wouldn’t the authors like to conduct network meta-analysis, and in particular IPD network meta-analysis, to compare different agents or classes of agents, and also to identify subgroups particularly suited or not-suited for particular agents? MINOR POINTS
--

	[ ] I don't seem to find the dates of the study, as required by the policy of the journal. [ ] Perhaps the authors need to report the checklists for PRISMA-IPD as well?
--	---

VERSION 1 – AUTHOR RESPONSE

Reviewer(s)' Comments to Author:

Reviewer: 1

Reviewer Name: Suzanne Oparil

This manuscript describes the impressive recent activity of the Blood Pressure Lowering Treatment Trialists' Collaboration (BPLTTC) that aims to conduct individual patient-level data (IPD) meta-analyses of published BP lowering randomized controlled trials (RCTs) in order to further enhance our understanding of the efficacy and safety of BP lowering treatment. The investigators have already made substantial progress in obtaining data from 49 trials with over 315,000 participants in all. The novelty of this approach is that it will yield sufficient IPD to assess the effects of BP lowering as a whole and also BP lowering with specific drug classes on a large variety of clinical outcomes, including, in addition to major cardiovascular events, new-onset diabetes, peripheral vascular disease, atrial fibrillation, fractures, cancer, cognitive function decline and vascular dementia. The adverse effects of antihypertensive treatment are less well appreciated than the positive effects of BP lowering based on individual trial results; the proposed analysis will fill in the gaps unsatisfied by individual trial results and will have major impact on individual patient management, clinical decision making and antihypertensive treatment guidelines. The organization of the investigation and the quality of the investigators are first rate. There are no major weaknesses in the proposal.

Many thanks for appreciating the importance of the research aims and study design of the BPLTTC.

Reviewer: 2

Reviewer Name: Toshi A. Furukawa

The authors present an overall protocol (not a specific protocol or statistical analysis plan) of their third round of IPD meta-analyses of blood pressure lowering drugs. They present the study selection procedure (which has already been completed), the data collection and management procedures (which are ongoing), and list their three broad categories of primary outcomes, but they do not present detailed statistical analyses. Instead they write "For each of these objectives, at least one study protocol will be developed."

The author team has a long tradition of conducting meaningful analyses based on IPD and I have only minor comments.

MINOR BUT ESSENTIAL POINTS

Could the authors present more details about the ethical and legal issues involved in collecting, managing and analyzing IPD?

Details about the governance, process of requesting data from data providers including information on the collection, handling, storage and analysis of the data are described in the protocol (page 7-9). We do not see any ethical issue (and we have obtained approval from the Institutional Review Board as described in the text on page 10, last paragraph). Data providers and the University of Oxford sign up to a Data Transfer Agreement which describes the conditions associated with the data transfer, storage and use by the Collaboration.

The authors write, "As we do not have access to patient identifiable information, our procedures are fully compliant with the Data Protection Act 1998." But how about the newer regulations such as EU's GDPR? And how about obtaining IPD from non-EU countries? Do they present particular difficulties?

The data that we receive from our collaborators are 'anonymised', therefore, not subject to GDPR (<https://gdpr-info.eu/>). The Data Transfer Agreement we have put in place required data providers to ensure that the data they share with us are anonymised and that the BPLTTC/analytic team at Oxford will not attempt to trace or identify any study participant. In this agreement, the data provider 'warrants that data have been obtained with the full consent and knowledge of the individual, including in respect of the proposed use and processing under and as envisaged by the agreement.' It is highly improbable that we can identify patients from the datasets we receive as we do not have access to the original data or linkage code.

We apply the same principles in our Data Transfer Agreement with investigators/data providers across EU and non-EU trials. We have not seen any difficulty in involving non-EU trials into the Collaboration. The more relevant issue usually involves tracing investigators or data custodians who are either deceased or whose contact details (e.g. email address no longer works) have changed. If the reviewer could put us in touch with some of the investigators or data custodians of trials whom we could not trace, we would greatly appreciate it.

The authors also write, "The University of Oxford's research contracts team is well aware of legal requirements for data sharing and storage and all our procedures are fully compliant with these requirements." What are "these requirements"? The readers would appreciate more details and difficulties involved in these processes.

The conduct of research at University of Oxford involves compliance to code of practice and procedures (<https://researchsupport.admin.ox.ac.uk/governance/integrity>) and to policies around research data handling and management (<http://researchdata.ox.ac.uk/>). To obtain further details about these policies and procedures, we now have added these website addresses in the text (page 10, last paragraph). Some difficulties we encounter would involve data providers or custodians who are not willing to share data with us.

The title should reflect the fact that the authors do not report details of analyses for specific outcomes but intend to write separate protocols for them. Otherwise the readers might assume that paper is such a protocol (as is usually the case for a systematic review protocol).

Probably "An overall protocol" instead of simply "A protocol"?

Many thanks for this suggestion. We have now modified the title as suggested to reflect the nature of this research protocol.

In COI statement, the authors write, "George Health Enterprises, the social enterprise arm of The George Institute, has received investment to develop fixed dose combinations of aspirin, statin and blood pressure lowering drugs." Which "blood pressure lowering drugs"?

These drugs are a combination of generic blood pressure-lowering agents. We also updated and expanded the COI statement to reflect current status: “George Health Enterprises has submitted patents for low-dose blood pressure combinations on which AR is listed as one of the inventors. AR does not have a financial interest in these planned products.”

MINOR AND DISCRETIONARY POINTS

I would assume that the included trials would involve many different drugs. Wouldn't the authors like to conduct network meta-analysis, and in particular IPD network meta-analysis, to compare different agents or classes of agents, and also to identify subgroups particularly suited or not-suited for particular agents?

This suggestion is very helpful. Network meta-analyses have largely been applied on aggregate data, but there are indeed methods for conducting IPD network meta-analyses. We will consider this approach as one of the possible analytical tools we could use. We now have mentioned this in the statistical analysis section.

MINOR POINTS

I don't seem to find the dates of the study, as required by the policy of the journal.

If the reviewer meant the dates of trials that we will consider in the meta-analyses, we are covering all studies from 1 January 1966 to 1 June 2018 (Page 6, middle paragraph). The conduct of the study, which has already begun, is dependent on available resources to achieve the proposed investigations. We therefore could not define when the current phase of the Collaboration will finish. Moreover, members of the Collaboration could propose new research projects which are beyond the scope the current project. In any case, each of these proposed investigations will require development and submission of a protocol with proposed timelines to aim for.

Perhaps the authors need to report the checklists for PRISMA-IPD as well

We plan to report PRISMA-IPD when reporting the protocol for each planned investigation.

VERSION 2 – REVIEW

REVIEWER	Toshi Furukawa Kyoto University, Japan TAF reports personal fees from Meiji, Mitsubishi-Tanabe, MSD and Pfizer and a grant from Mitsubishi-Tanabe; TAF has a patent 2018-177688 pending.
REVIEW RETURNED	05-Mar-2019

GENERAL COMMENTS	The authors have responded adequately to all my comments.
---